# Efflux Pump (QacA, QacB, and QacC) and β-Lactamase Inhibitors? An Evaluation of 1,8-Naphthyridines against *Staphylococcus aureus* Strains

**DOI:** 10.3390/molecules28041819

**Published:** 2023-02-15

**Authors:** Cícera Datiane de Morais Oliveira-Tintino, Saulo Relison Tintino, Ana Carolina Justino de Araújo, Cristina Rodrigues dos Santos Barbosa, Priscilla Ramos Freitas, José Bezerra de Araújo Neto, Iêda Maria Begnini, Ricardo Andrade Rebelo, Luiz Everson da Silva, Sandro Lucio Mireski, Michele Caroline Nasato, Maria Isabel Lacowicz Krautler, Humberto Medeiros Barreto, Jaime Ribeiro-Filho, Irwin Rose Alencar de Menezes, Henrique Douglas Melo Coutinho

**Affiliations:** 1Laboratory of Microbiology and Molecular Biology, Department of Biological Chemistry, Regional University of Cariri (URCA), Crato 63105-000, CE, Brazil; 2Department of Chemistry, Regional University of Blumenau (FURB), Itoupava Seca, Blumenau 89030-903, SC, Brazil; 3Postgraduate Program in Sustainable Territorial Development, Coastal Sector, Federal University of Paraná (UFPR), Curitiba 81531-990, PR, Brazil; 4Laboratory of Microbiology, Federal University of Piaui (UFPI), Teresina 64049-550, PI, Brazil; 5Oswaldo Cruz Foundation (Fiocruz), Fiocruz Ceará, Eusébio 60180-900, CE, Brazil; 6Laboratory of Pharmacology and Molecular Chemistry (LFQM), Department of Biological Chemistry, Regional University of Cariri (URCA), Crato 63105-000, CE, Brazil

**Keywords:** 1,8-naphthyridines, bacterial resistance, β-lactamase, QacA/B efflux pumps, *S. aureus*

## Abstract

The bacterial species *Staphylococcus aureus* presents a variety of resistance mechanisms, among which the expression of β-lactamases and efflux pumps stand out for providing a significant degree of resistance to clinically relevant antibiotics. The 1,8-naphthyridines are nitrogen heterocycles with a broad spectrum of biological activities and, as such, are promising research targets. However, the potential roles of these compounds on bacterial resistance management remain to be better investigated. Therefore, the present study evaluated the antibacterial activity of 1,8-naphthyridine sulfonamides, addressing their ability to act as inhibitors of β-lactamases and efflux pump (QacA/B and QacC) against the strains SA-K4414 and SA-K4100 of *S. aureus*. All substances were prepared at an initial concentration of 1024 μg/mL, and their minimum inhibitory concentrations (MIC) were determined by the broth microdilution method. Subsequently, their effects on β-lactamase- and efflux pump-mediated antibiotic resistance was evaluated from the reduction of the MIC of ethidium bromide (EtBr) and β-lactam antibiotics, respectively. The 1,8-naphthyridines did not present direct antibacterial activity against the strains SA-K4414 and SA-K4100 of *S. aureus*. On the other hand, when associated with antibiotics against both strains, the compounds reduced the MIC of EtBr and β-lactam antibiotics, suggesting that they may act by inhibiting β-lactamases and efflux pumps such as QacC and QacA/B. However, further research is required to elucidate the molecular mechanisms underlying these observed effects.

## 1. Introduction

*Staphylococcus aureus* is an opportunistic bacterium found mainly in the nasal cavity. Thus, while endogenous *S. aureus* strains are a frequent source of infection, strains acquired from external sources are also relevant in the context of infection [1,2]. *S. aureus* infection in superficial areas, such as soft tissues and the skin, can reach the bloodstream if natural barriers are broken. In this context, the bloodstream facilitates the spreading of the microorganism to other regions of mucosa and skin, causing severe generalized infections [3,4,5]. 

Antigenic molecules of the cell wall of *S. aureus*, such as protein A, teichoic acid, lipoteichoic acid, and adhesins, induce immune responses associated with the inflammatory process [6,7]. The pathogenic and host colonization potential of *S. aureus* is determined by the presence of virulence factors that are classified according to their function and stage of action. In this context, three categories of virulence factors stand out: factors that affect adhesion to the host, factors that inhibit the immune system, and factors that favor cell invasion [8].

In 2017, the World Health Organization (WHO) published a list of pathogens with research priority in which multidrug resistance (MDR) bacteria such as *S. aureus* are included [9]. The emergence of bacterial strains with an MDR phenotype can be explained by acquiring antibiotic-resistance genes through mutation or horizontal gene transfer (HGT) [10,11]. Over the years, *S. aureus* has acquired the ability to alter intrinsic physiological mechanisms to resist antibiotics. The ultimate source of mutation in the evolutionary process is initiated by recurrent exposure to antimicrobial agents, which contributes to the development of biochemical and genetic alterations observed in resistant strains [12,13]. 

Consistent evidence has pointed to β-lactamase and efflux pumps as significant molecular targets in the search for new antibacterial compounds [14]. In this context, combination therapy is a critical alternative with significant benefits to pharmacotherapy. This strategy assumes that combining synthetic substances with antibiotics can affect many therapeutic targets and result in synergistic interactions that potentiate the antibacterial properties with less susceptibility to microbial adaptation [15].

The β-lactamase family of enzymes degrades β-lactam antibiotics by mediating the hydrolysis of the amide bond of the four-membered β-lactam ring, conferring thus bacterial antibiotic resistance in Gram-positive and Gram-negative bacteria [16,17]. Unfortunately, in the fight against β-lactamase-mediated resistance, the diversification of antibiotics and using derivatives with improved action stimulated the emergence of hydrolytic enzymes with extended action spectra. Thus, while β-lactamase inhibitors were effective against first-generation β-lactams, the rise of extended-spectrum β-lactamases (ESBLs) has resulted in resistance to all antibiotics in this class [18,19]. The spread of β-lactamase genes has been significantly favored by mobile genetic elements, such as plasmids or transposons, which facilitate the rapid transfer of genes from one bacteria to another [16]. Notably, evidence has shown that strains such as the QaC present a linkage between the genes of β-lactamases and efflux pumps [20,21].

Efflux pumps carry several transmembrane domains that form a channel for the passage of toxic compounds, such as antibiotics, cations, compounds of quaternary ammonium, phosphonium derivatives, and DNA intercalating dyes, which are extruded from the bacterium. Notably, several families of efflux pumps have been identified in multidrug-resistant *S. aureus*, mediating antibiotic resistance in this Gram-positive microorganism [22,23,24].

Efflux pump-encoding plasmids have been consistently identified in *S. aureus* strains, potentially contributing to their reduced susceptibility to antiseptics and disinfectants, such as cetrimide, benzalkonium chloride, and chlorhexidine [22,25]. Evidence has indicated that these genes have evolved to confer increasing specificity to substrates, protecting against harmful agents, including various antibiotics. In clinically relevant populations, these genes may have been selected due to the human use of disinfectants [26,27], although there is no consistent experimental evidence of this phenomenon. [28]. Among the products of these genes is a group of proteins collectively known as QaC for quaternary ammonium compound resistance/transporter [29].

The QaC efflux proteins expressed by *Staphylococcus* species belong to two families: the major facilitator superfamily (MFS) and the small multidrug resistance (SMR) family [30]. The QacA protein is encoded by plasmids that confer resistance to several distinct substrates, such as quaternary ammonium compounds, dyes, DNA intercalants, and antibiotics. When the gene encoding QacB was sequenced from plasmid pSK23, it was found to be very similar to *qacA*, despite the observed difference in the substrate that had been the basis of their distinction [29,31,32,33]. 

Some efflux proteins have evolved over the years by modifying external structures such as transmembrane segment (TMS) domains. Proteins such as QacA and QacB have undergone this modification process reaching 14 TMS that presents two additional domains located in the center, which are the products of intragenic duplication. The QacC protein was first recognized in the circular plasmid pSk89 of an *S. aureus* strain. This protein is composed of 107 amino acids, although slight variation can occur when this protein is translated into another bacterial genus [34,35].

Given the importance of these proteins in antibiotic resistance and bacterial metabolic functions, the identification of efflux pump inhibitors (EPIs) has been the target of several drug development studies [32,33]. These molecules are administered along with antibiotics to prevent their extrusion and increase their intracellular concentration [36,37]. Therefore, in addition to restoring antibiotic susceptibility and enhancing its action, EPIs can reduce the occurrence of persistent strains, helping to treat chronic infections [38,39].

Literature data show that synthetic compounds from different classes have promising characteristics in the search for new antibacterial agents with the potential to reduce antimicrobial resistance [40]. The naphthyridines, also known as pyridopyrines, are heterocyclic compounds presenting a nitrogen atom in each ring. Naphthyridines, or diazanaphthalenes, are synthetic compounds with varying structures. The first naphthyridine derivative was synthesized in 1893 by Reissert, who identified naphthalene as a pyridine analog [41]. The first unsubstituted naphthyridines, 1,5-naphthyridine and 1,8-naphthyridine were prepared in 1927 [42,43,44]. Among the isomers presented, 1,8-naphthyridine derivatives show the most promising biological and chemical properties, reflecting the significant number of publications and patents [45,46]. This class of compounds is attractive in the context of drug development research since their chemical and biological activities have been attested by an increasing number of studies, with the antibacterial activity being the most explored among the biological effect of this class [47,48,49,50,51]. 

The history of research on the biological activity of 1,8-naphthyridines initiated with the synthesis of nalidixic acid and the demonstration of its antibacterial properties against Gram-negative bacteria causing urinary tract infections [52,53]. Notably, nalidixic acid is a synthetic quinolone discovered by Lesher and collaborators as a distilled by-product during antimalarial chloroquine synthesis [52]. Because the literature exploring the theme is quite extensive, 1, 8-naphthyridine derivatives can be grouped into four categories according to their chemical properties: azetidinylquinolones, pyridone-thiazole, and carboxylic acids [48].

E-4695, an azetidinylquinolone derivative, is a fluorinated naphthyridine with an azetidine fraction exhibiting promising antibacterial properties against Gram-positive cocci of the genera *Streptococcus*, *Staphylococcus,* and *Enterococcus*. Previous research has suggested that this compound is more effective than ofloxacin and ciprofloxacin against some Gram-negative bacterial strains, including strains of the Enterobacteriaceae family, *Bacteroides fragilis*, and *Clostridium perfringens*. In addition, this compound was found to present promising pharmacological effects in an in vivo model of *S. pneumoniae* infection [54]. However, few studies have reported the antibacterial activity of naphthyridines and their sulfonamide derivatives. 

Therefore, the present study evaluated the antibacterial activity of 1,8-naphthyridine sulfonamides, addressing their ability to act as inhibitors of β-lactamases and efflux pumps in *Staphylococcus aureus* strains.

## 2. Results and Discussion

### 2.1. Antibacterial Activity of 1,8-Naphthyridines

The antibacterial activity analysis revealed that none of the 1,8-naphthyridines presented clinically relevant MICs against *S. aureus* strains SA-4100 and SA-K4414. If, on the one hand, the absence of antibacterial activity discourages using these compounds to treat infections caused by these *S. aureus* strains; on the other hand, it does not limit the potential use of 1,8-naphthyridines as efflux pump inhibitors since the absence of an antibacterial effect prevents the subsequent development of resistance [55], besides avoiding the overlapping of antibacterial effects [56]. In addition, despite the above-described results, evidence has shown that some naphthyridines have antibacterial activity against *S. aureus* and other MDR pathogenic bacteria [57,58,59]. Moreover, it is crucial to emphasize that naphthyridines are precursors of antibiotics such as quinolones and fluoroquinolones [60,61,62,63,64], which encourages the development of research targeting the use of these substances as antibiotics and antibiotic enhancers.

In addition to the 1,8-naphthyridines, sulfonamides are an important class of substances in the drug development research context. Previous research has demonstrated that sulfonamides are highly efficient antibacterial compounds, although the chemical structure and mechanisms of action differ from 1,8-naphthyridines. Sulfonamides are the oldest class of synthetic antibiotics, with proven efficacy against many microbial infections and wide use in clinical practice [65,66]. These broad-spectrum antibacterial compounds demonstrated significant effectiveness in treating acute and chronic infections caused by Gram-positive and Gram-negative strains and have the advantage of do not affect the host’s defense mechanisms [67,68]. In addition, studies confirmed that these compounds have a bacteriostatic action, with sulfanilamide being the active part of the molecule [69,70].

### 2.2. Confirmation of β-Lactamase Activity in the SA-K4414 and SA-4100 Strains of S. aureus

As shown in Table 1, the association with sulbactam resulted in a 3-fold reduction of ampicillin MIC against the strain SA-K4414 compared to the controls treated exclusively with ampicillin. Regarding the SA-K4100, the combined treatment caused a 4-fold reduction in ampicillin MIC. On the other hand, the addition of CCCP has no significant impact on the MIC of ampicillin or ampicillin + sulbactam against SA-K4414 or SA-4100, indicating that both strains present β-lactamase activity that is not influenced by efflux pump inhibition with CCCP.

Previous research has identified a relationship between the presence of genes of resistance to quaternary compounds, which confer resistance through QacA, QacB, and QacC efflux pumps, and genes of β-lactamase, which confer resistance to β-lactams. In fact, the genes *qacA/B* and *blaZ* are commonly found in the same bacterial plasmid and are associated with mobile genetic elements [20,21,35,71,72,73,74]

It is known that the strains SA-K4414 and SA-K4100 of *S. aureus* express efflux pump [75] and β-lactamase-mediated resistance mechanisms. Our findings suggest, however, that the QacA/B and QacC efflux pumps, expressed respectively by SA-K4414 and SA-K4100, do not interfere with the activity of β-lactams, considering that the association with these antibiotics caused no significant change in the MIC. 

### 2.3. Evidence of β-Lactamase Inhibition by 1,8-Naphthyridines

As demonstrated in Figure 1A,B, all naphthyridines reduced the MICs of the antibiotics tested against the strains SA-K4414 and SA-K4100 of *S. aureus*. In SA-K4414 cultures, Naph 1 and Naph 3 reduced the MIC of penicillin by caused 4-fold and 8-fold, respectively, probably due to β-lactamase inhibition. Comparable findings were observed with Naph 1, Naph 3, and Naph 4, which reduced the MIC of oxacillin against the SA-K4100 strain by 5-, 4-, and 8-fold, respectively [76]. 

Clinically available β-lactamase inhibitors, such as sulbactam, clavulanic acid, and tazobactam, have significantly lost their antibacterial efficacy in the current antibiotic resistance scenario. Each of these three inhibitors shares a β-lactam backbone that interacts with its target in the bacterium. However, given the high speed with which β-lactamases evolve to overcome the action of these drugs, the development of new β-lactamase inhibitors is urgently demanded [77,78].

The literature reports some molecules that are capable of inhibiting β-lactamase activity, including diazabicyclooctanes [79] PA-34 (based on acylated phenoxyaniline) [80], cyclic boronates [81], and avibactam [82], which have larger structures than the clinically available inhibitors. However, despite the structural differences, many new inhibitors, like the naphthyridines, have one or more cyclic structures in their composition [83]. 

This pioneering study evaluated the role of naphthyridines as β-lactamase inhibitors. Since β-lactamases are synthesized and relocated on the outer side of the bacterial cell in Gram-positive bacteria (i.e., in the outer leaflet of the bacterial membrane), it is assumed that the penetration of naphthyridines into the bacterial cell is not necessary for their action [16]. While evidence suggests that naphthyridines can act as competitive inhibitors, binding to the beta-lactamase active site, further studies are needed to confirm this mechanism of action.

### 2.4. Evidence of Efflux Pump Inhibition by 1,8-Naphthyridines

The results shown in Figure 2A indicate that none of the evaluated naphthyridines reduced the MIC of EtBr against the SA-K4414 strain. On the other hand, Figure 2B shows that Naph 3 reduced the MIC of EtBr against the SA-K4100 strain by 8-fold, while the other compounds had no impact on the MIC of EtBr. As expected, the protonophore compound CCCP was able to reduce the MIC of the EtBr against both strains, indicating a MIC reduction associated with the inhibition of QacA/B and QacC efflux pumps. Accordingly, evidence indicates that compounds that are capable of reducing the MIC of the EtBr against efflux pump-expressing strains act as efflux pump inhibitors (EPI) [84,85,86,87,88].

An analysis of the figures described above reveals the uncontestable difference between Naph 3 and the other naphthyridines regarding their ability to inhibit efflux mechanisms in *S. aureus*. Such differential property of Naph 3 can be related to the presence of three fluorine substituent groups, which, due to the characteristic electronegativity, could have the binding to active sites on the efflux pump favored [89]. Evidence indicates that fluoroquinolone derivatives presenting fluorine groups and nitrogen atoms linked to the aromatic rings can act as efflux pump inhibitors, supporting our hypothesis [90].

The Qac efflux pumps can transport a variety of compounds, including quaternary compounds, toxic metals, and EtBr, which penetrate the bacterial cytoplasm causing toxic effects on the bacterial DNA [29]. Efflux pumps can extrude their substrates through efficient antiport, uniport, or symport transports. Examples of QacA’s specific substrates are ethidium bromide, cetrimide, pentamidine, benzalkonium, and chlorhexidine [91,92,93]. On the other hand, the QacB pump is responsible for extruding ethidium bromide, acriflavine, benzalkonium chloride, tetraphenylphosphonium, and rhodamine [94], while the QacC pump exports ethidium bromide cetrimide, benzalkonium chloride, chlorhexidine diacetate [95]. Therefore, it is possible to affirm that the bacterial strains that express these efflux pumps present high resistance to EtBr, using it as a substrate [96]. 

The inhibition of efflux mechanisms makes bacteria more susceptible to antibacterial agents, as these compounds can more easily accumulate in the intracellular medium [97]. In addition, efflux pump inhibition can affect critical physiological processes such as osmotic balance, pH maintenance, nutrient acquiring, and metabolite expelling metabolic [98,99]. 

Naphthyridines and their derivatives are known precursors of antibiotics, such as fluoroquinolones and quinolones [61,62,64,100]. The efflux pump inhibitory capacity of these substances is favored by their high binding affinity to these proteins, as verified for 2,3,4-trifluoro-N-(5-chloro-1,8-naphthyridin-2-yl)-benzenesulfonamide (Naph 3), which showed favorable interaction with NorA and MepA proteins in silico [101]. Previous research also demonstrated that the naphthyridines tested in the present study (Naph 1-5) have the potential to inhibit the NorA, MepA, Tet(K), and MsrA proteins, corroborating the present data [101,102,103]. These results were obtained by evaluating the fluorescence of EtBr, the reduction of the MIC of antibiotics, and by in silico molecular docking analysis [101,103]. Other studies have shown the inhibition of NorA by 1,8-naphthyridines in EtBr efflux inhibition assays and using in silico analysis of a focused scaffold hopping approach with a virtual pharmacophore screening [104].

Therefore, it has been suggested that the potential mechanism of action underlying the antibiotic-enhancing activity of Naphs are the following: competitive or non-competitive inhibition of the antibiotic binding into the efflux pump and lactamase active site; direct interaction with β-lactamases; inhibition of efflux pump gene expression; and formation of Naph-antibiotic complexes, preventing the extrusion of the antibiotic or the breakdown of the antibiotic by β-lactamase. However, this hypothesis needs to be better investigated [105].

## 3. Materials and Methods

### 3.1. Drugs and Reagents

The efflux pump substrates, ethidium bromide (EtBr) and carbonyl cyanide *m*-chlorophenylhydrazine (CCCP), and the antibiotics oxacillin, ampicillin/sulbactam, and penicillin, were obtained from Sigma-Aldrich Co. Ltd, San Luis, MO, USA. The 1,8-naphthyridine sulfonamides and 1,8-naphthyridinone used in the study are listed in Table 2. All compounds were synthesized and provided by Luiz Everson da Silva, Ph.D., from the Federal University of Paraná, and by researchers Iêda Maria Begnini, Ph.D., Ricardo Andrade Rebelo, Ph.D., Sandro Lucio Mireski, Ph.D., Michele Caroline Nasato, M.Sc. and Maria Isabel Lacowicz Krautler, B.ASc., from the Regional University of Blumenau. The synthesis and full characterization of the listed compounds have been previously described in the literature [102]. The antibiotics and heterocycles were dissolved in dimethyl sulfoxide (DMSO) and diluted in sterile distilled water to a concentration of 1024 μg/mL. Compounds CCCP and EtBr were diluted directly in sterile distilled water to 1024 μg/mL.

### 3.2. Microorganisms

The strains SA-K4414 and SA-K4100 of *S. aureus* were provided by Prof. Glenn Kaatz (Wayne State University). These bacterial strains carry the genes *QacA/B* and *QacC*, respectively, which encode the QacA/B and QacC efflux pumps (MFS and SMR protein family), beside orchestrating the production of β-lactamases [31]. Twenty-four hours before the experiments, the bacterial strains were sown in a heart infusion agar (HIA, Difco Laboratories Ltd, Detroit, MI, USA) culture medium and incubated at 37 °C.

### 3.3. Minimum Inhibitory Concentration (MIC) Determination and Antibacterial Analysis

The antibacterial activity was evaluated using the broth microdilution method. Eppendorf® tubes were added with 100 μL of bacterial inoculum suspended in saline solution and 900 μL of brain heart infusion (BHI) culture medium. Then, 100 μL of this solution was transferred to the wells on a 96-well plate, followed by the addition of 100 μL of the solution of the compounds at concentrations ranging from 512 to 0.5 μg/mL (1:1) by serial dilution. The plates were incubated at 37 ºC for 24 h, and then resazurin (7-hydroxy-3H-phenoxazin-3-one 10-oxide) was added to perform the readings. The MIC was defined with the lowest concentration, ranging from 512 μg/mL to 0.5 μg/mL, capable of inhibiting bacterial growth [76]. 

### 3.4. Analysis of β-Lactamase Activity

To analyze β-lactamase activity, we determined the MIC of sodium ampicillin alone and combined it with the β-lactamase inhibitor, sodium sulbactam, at a concentration of 50%. Enzymatic inhibition was confirmed when the enzyme inhibitor caused a 3-fold reduction in the antibiotic MIC [76]. In addition, to exclude the participation of an efflux pump-dependent mechanism in this assay, the MIC of ampicillin and ampicillin + sulbactam was determined in the presence of an efflux pump inhibitor used at a subinhibitory concentration (MIC/8).

### 3.5. Verification of β-Lactamase and Efflux Pump Inhibition

To verify if 1,8-naphthyridines could act as β-lactamase and efflux pump inhibitors, we evaluated whether these substances could reduce the MIC of β-lactam antibiotics and EtBr, respectively. The antibiotics chosen for this inhibition assay are specific efflux [106,107]. Of note, EtBr is a substrate for efflux pumps in MDR bacteria.

#### 3.5.1. Evaluation of β-Lactamase Inhibition through Reduction of Antibiotic MIC 

This analysis was performed by evaluating the impact of the 1,8-naphthyridines on the MIC of penicillin and oxacillin. Eppendorf tubes containing the compounds in volumes corresponding to their MIC/8 were added with 150 μL of bacterial inoculum and 10% BHI to complete a final volume of 1500 μL. Positive controls were prepared with 150 μL of inoculum and 1350 μL 10% BHI. After preparation, the solutions were transferred to the wells on a 96-well plate, followed by the addition of 100 μL of antibiotics at variable concentrations obtained by serial dilution (1:1). The plates were incubated at 37 °C for 24 h, and bacterial growth was evaluated after the addition of resazurin. The MIC was defined as the lowest antibiotic concentration, ranging from 512 μg/mL to 0.5 μg/mL, capable of inhibiting bacterial growth.

#### 3.5.2. Evaluation of Efflux Pump Inhibition through the Reduction of Ethidium Bromide MIC

This experiment evaluated whether 1,8-naphthyridines could decrease the MIC of the efflux pump substrate EtBr, indicating efflux pump inhibition. Briefly, the 1,8-naphthyridines bacterial inocula and media were added to the wells as previously described (Section 3.5.1). Subsequently, the plates were filled with 100 μL of EtBr at concentrations ranging between 512 μg/mL and 0.5 μg/mL through serial dilution (1:1). The plates were incubated at 37 ºC for 24 h, and the readings were performed as previously described. The MIC of controls was assessed using EtBr alone without adding the test compounds.

### 3.6. Statistical Analysis of Microbiological Results

The experiments were performed in triplicates, and the results were expressed as the means of the replicates. The results of the tests are expressed as the geometric mean. Statistical hypothesis analysis was applied using a two-way ANOVA followed by a Bonferroni’s post hoc test, using the GraphPad Software 5.0 software, San Diego, CA, USA. Differences with a *p* < 0.05 were considered significant.

## 4. Conclusions

Although 1,8- naphthyridines do not have intrinsic antibacterial activity against *S. aureus* strains, they can potentiate the action of antibiotics, probably by interfering with bacterial resistance targets such as β-lactamase and the QacAB and QacC efflux pumps and therefore have potential to be used antibiotic adjuvants. However, further research is required to elucidate the mechanisms underlying the inhibition of these targets 1,8- naphthyridines and evaluate their toxicity to eukaryotic cells. 

Regarding the mechanisms involved in antibiotic resistance modulation, molecular biology analysis will provide more consistent evidence on the interaction of 1,8- naphthyridines with efflux pumps, β-lactamases, and other potential molecular targets in the context of antibiotic resistance in *S. aureus*.

## Figures and Tables

**Figure 1 molecules-28-01819-f001:**
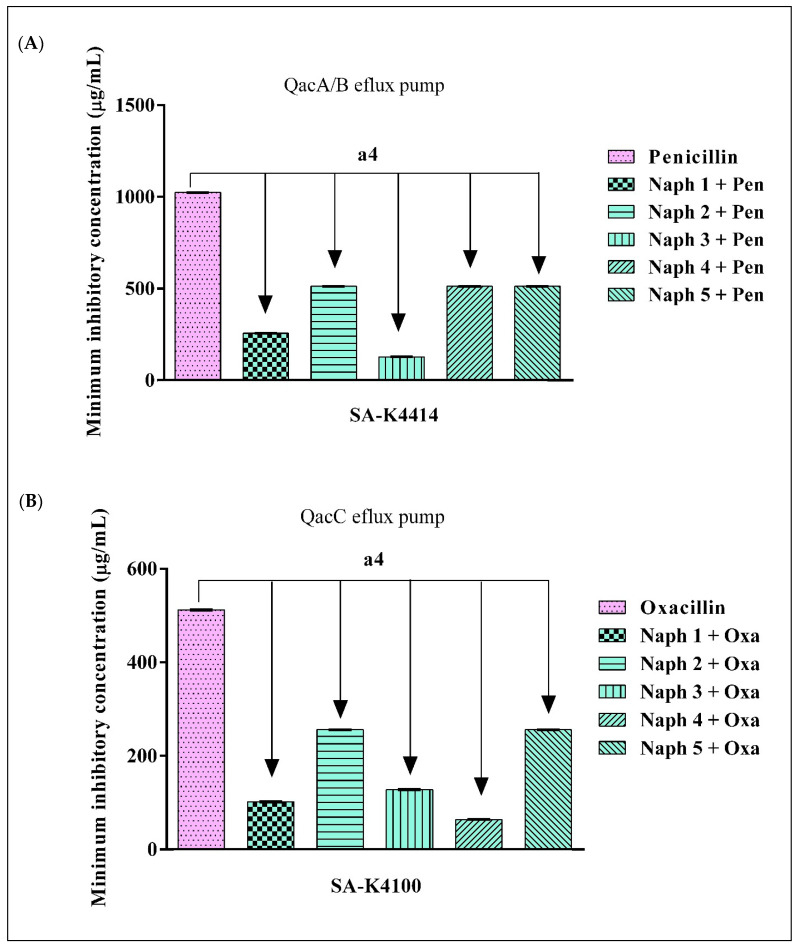
Inhibition of β-lactamase by the 1,8-naphthyridines associated with penicillin and oxacillin against the (**A**) SA-K4414 that presenting QacA/B efflux pump and (**B**) SA-K4100 that presenting QacC efflux pump. Both strains that present the β-lactamase enzymatic resistance mechanism. The values represent the geometric mean ± S.E.M. (standard error of the mean). A two-way ANOVA, followed by the Bonferroni test was used for analysis. a4 = *p* < 0.0001 vs. control; Naph = Naphthyridine; Oxa = Oxacillin; Pen = Penicillin.

**Figure 2 molecules-28-01819-f002:**
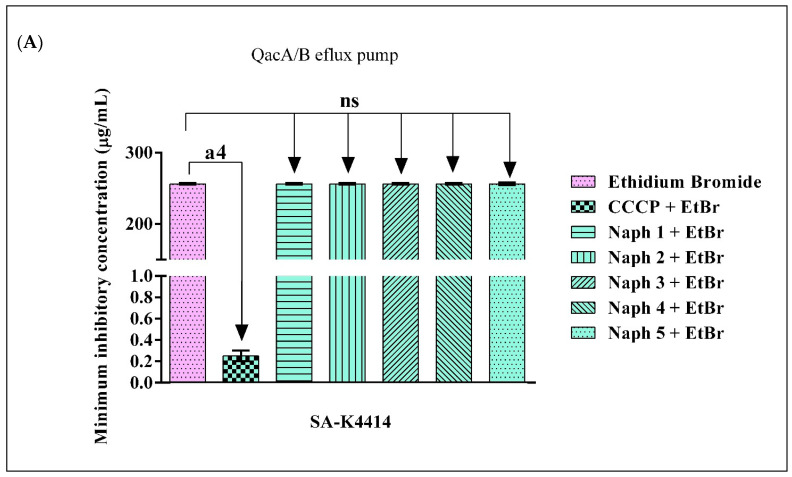
Evaluation of the inhibitory effect of 1,8-naphthyridines and CCCP, associated with ethidium bromide against the SA-K4414 contain QacA/B efflux pumps (**A**), and SA-K4100 strains with QacC efflux pumps (**B**). The values represent the geometric mean ± S.E.M. (standard error of the mean). A two-way ANOVA, followed by the Bonferroni test was used for analysis. a4 = *p* < 0.0001 vs. control; CCCP = Carbonyl cyanide m-chlorophenylhydrazine; EtBr = Ethidium bromide; Naph = Naphthyridine; ns = Not significant.

**Table 1 molecules-28-01819-t001:** Minimum Inhibitory Concentration of Ampicillin in combination with β-lactamase and Efflux Pump Substrates.

Treatment	SA-K4414	SA-K4100
Ampicillin	≥1024 μg/mL	128 μg/mL
Ampicillin + Sulbactam	128 μg/mL	16 μg/mL
Ampicillin + CCCP (MIC/8)	≥1024 μg/mL	128 μg/mL
Ampicillin + Sulbactam + CCCP (MIC/8)	128 μg/mL	16 μg/mL

**Table 2 molecules-28-01819-t002:** Structure of the 1,8-naphthyridines and 1,8-naphthyridinone used.

Compound	Structure	Compound Name	Yield%	Melting Point°C
Naph 1	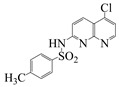	4-methyl-*N*-(5-chloro-1,8-naphthyridin-2-yl)- benzenesulfonamide	84	249–250
Naph 2	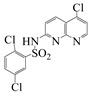	2.5-dichloro-*N*-(5-chloro-1,8-naphthyridin-2-yl)-benzenesulfonamide	70.8	247.5–248
Naph 3	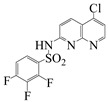	2,3,4-trifluoro-*N*-(5- chloro-1,8-naphthyridin-2-yl)-benzenesulfonamide	56.5	198.6–199.6
Naph 4	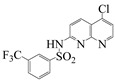	3-trifluoromethyl-*N*-(5-chloro-1,8-naphthyridin-2-yl)-benzenesulfonamide	52.6	222.2–224
1,8-naphthyridinone	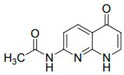	7-acetamido-1,8-naphthyridin-4(*1H*)-one	-	298–299

## Data Availability

The data are available upon official request from the principal investigator.

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
