# Peer review of "Efflux Pump (QacA, QacB, and QacC) and β-Lactamase Inhibitors? An Evaluation of 1,8-Naphthyridines against *Staphylococcus aureus* Strains"

_molecules, 2023, doi:10.3390/molecules28041819_

Round 1
Reviewer 1 Report
Text must be revised, below some misspells, errors, and observations, etc.
Line 46. Punctuation in “…[1,2]. Over the..”
Line 58. Delete word “which”, change to “…through toxic compounds,…”
Line 100. Delete doubled word “However”
Line 129. Word misspelling “…pum…” change to “…pump…”
Figure 1 and 2. Values (bars) of standard error of mean did not appear or maybe they were not calculated properly. Please check the mathematics to add the bars correctly.
Table 2. In column “Compound name” only -N- and (1H)- must be in italics. Check if “acestarch….” Should be in lower case as the others.
Table 2. Column “Compound” Naph 3 is missing, while Naph 2 is duplicated
Section 3.1. Line 219-234, there are no references about the synthesis of Naph 1-4 and 1.8-naphthyridinone; it is important to indicate purity of compounds, at least elemental analysis (%C, %N) and melting temperature should be mentioned.
Author Response
We want to thank you for giving us a chance to revise the manuscript again. Your comments and suggestions are invaluable in improving the quality of our manuscript. We revised the manuscript according to requests to minimize typographical, grammatical, and bibliographic errors and improve comprehension. The changes made to the revised manuscript were highlighted in the text.
Text must be revised, below some misspells, errors, and observations, etc.
Line 46. Punctuation in “…[1,2]. Over the..”
R=Done
Line 58. Delete word “which”, change to “…through toxic compounds,…”
R=Done
Line 100. Delete doubled word “However”
R=Done
Line 129. Word misspelling “…pum…” change to “…pump…”
R=Done
Figure 1 and 2. Values (bars) of standard error of mean did not appear or maybe they were not calculated properly. Please check the mathematics to add the bars correctly.
R= Dear reviewer, the standard error of the mean was between 1 and 0.58. The high values of the graph bars may have flattened the size of the error. But it is being represented on the chart by these darker lines above each bar.
Table 2. In column “Compound name” only -N- and (1H)- must be in italics. Check if “acestarch….” Should be in lower case as the others.
R= Done
Table 2. Column “Compound” Naph 3 is missing, while Naph 2 is duplicated
R= Done
Section 3.1. Line 219-234, there are no references about the synthesis of Naph 1-4 and 1.8-naphthyridinone; it is important to indicate purity of compounds, at least elemental analysis (%C, %N) and melting temperature should be mentioned.
R= Done

Reviewer 2 Report
The paper describes the antibacterial activity of 1,8-naphthyridine sulfonamides, addressing their ability to act as inhibitors of β-lactamases and efflux pump (QacA/B and QacC) inhibitors in the strains SA-K4414 and SA-K4100 of Staphylococcus aureus.
S. aureus. is one of highly virulent bacterial pathogens. It is part of ESKAPE pathogens. ESKAPE is an acronym for six highly virulent bacterial pathogens including: Enterococcus faecium, Staphylococcus aureus, Klebsiella pneumoniae, Acinetobacter baumannii, Pseudomonas aeruginosa, and Enterobacter cloacea. These pathogens are multidrug resistant and are the main causative agents of infections, including nosocomial infections.
The 1,8-naphthyridines did not present direct antibacterial activity against the strains SA-K4414 and SA-K4100 of S. aureus. However, when they associated with antibiotics against both strains, the compounds reduced the MIC of EtBr and 36 β-lactam antibiotics, suggesting that they may act by inhibiting β-lactamases and efflux such as 37 QacC and QacA/B efflux pumps.
The manuscript is written clearly and understandably without frills. All conclusions supported by the results. The paper can be published in Molecules after minor revision.
1. Why the authors limited their research to S. aureus.?
2. The conclusions are rather brief and need to be expanded.
Author Response
We want to thank you for giving us a chance to revise the manuscript again. Your comments and suggestions are invaluable in improving the quality of our manuscript. We revised the manuscript according to requests to minimize typographical, grammatical, and bibliographic errors and improve comprehension. The changes made to the revised manuscript were highlighted in the text.
The paper describes the antibacterial activity of 1,8-naphthyridine sulfonamides, addressing their ability to act as inhibitors of β-lactamases and efflux pump (QacA/B and QacC) inhibitors in the strains SA-K4414 and SA-K4100 of Staphylococcus aureus.
- aureus.is one of highly virulent bacterial pathogens. It is part of ESKAPE pathogens. ESKAPE is an acronym for six highly virulent bacterial pathogens including: Enterococcus faecium, Staphylococcus aureus, Klebsiella pneumoniae, Acinetobacter baumannii, Pseudomonas aeruginosa, and Enterobacter cloacea. These pathogens are multidrug resistant and are the main causative agents of infections, including nosocomial infections.
The 1,8-naphthyridines did not present direct antibacterial activity against the strains SA-K4414 and SA-K4100 of S. aureus. However, when they associated with antibiotics against both strains, the compounds reduced the MIC of EtBr and 36 β-lactam antibiotics, suggesting that they may act by inhibiting β-lactamases and efflux such as 37 QacC and QacA/B efflux pumps.
The manuscript is written clearly and understandably without frills. All conclusions supported by the results. The paper can be published in Molecules after minor revision.
1.Why the authors limited their research to S. aureus.?
R = Dear reviewer, first of all, we thank you for your considerations. We chose S. aureus strains for two reasons: first, in this strain the efflux pumps have already been identified. The second reason was that, according to the literature, the genes for Qacc and QacA/B pumps are generally found on the same plasmid as the beta-lactamse genes. Although nothing prevents further studies to evaluate the action on other species. So we appreciate your understanding.
- The conclusions are rather brief and need to be expanded.
R = Done
